# Percutaneous Computed Tomography-Guided Coaxial Core Biopsy for the Diagnosis of Pancreatic Tumors

**DOI:** 10.3390/jcm8101633

**Published:** 2019-10-05

**Authors:** Yung-Yeh Su, Yi-Sheng Liu, Ying-Jui Chao, Nai-Jung Chiang, Chia-Jui Yen, Hong-Ming Tsai

**Affiliations:** 1National Institute of Cancer Research, National Health Research Institutes, Tainan 704, Taiwan; yysu@nhri.edu.tw (Y.-Y.S.); njchiang@nhri.org.tw (N.-J.C.); 2Division of Hematology and Oncology, Department of Internal Medicine, National Cheng Kung University Hospital, College of Medicine, National Cheng Kung University, Tainan 704, Taiwan; yencj@mail.ncku.edu.tw; 3Institute of Clinical Medicine, College of Medicine, National Cheng Kung University, Tainan 704, Taiwan; 4Department of Diagnostic Radiology, National Cheng Kung University Hospital, College of Medicine, National Cheng Kung University, Tainan 704, Taiwan; taicheng100704@yahoo.com.tw; 5Department of Surgery, National Cheng Kung University Hospital, College of Medicine, National Cheng Kung University, Tainan 704, Taiwan; pitt_chao@yahoo.com.tw

**Keywords:** CT, coaxial, percutaneous biopsy, pancreatic tumor

## Abstract

Endoscopic, ultrasound-guided tissue acquisition (EUS-TA) with rapid on-site evaluation is recommended as a first choice in the diagnosis of pancreatic lesions. Since EUS facilities and rapid on-site evaluation are not widely available, even in medical centers, an alternative for precise diagnoses of pancreatic tumor is warranted. The percutaneous computed tomography-guided, core needle biopsy (CT-CNB) is a commonly applicable method for biopsies. Our institute has developed a fat-transversing approach for pancreatic biopsies which is able to approach most tumors in the pancreas without penetrating organs or vessels. Herein, we report a 15-year experiment of pancreatic tumor coaxial CT-CNB in 420 patients. The success rate of tissue yielding by the technique was 99.3%. The overall sensitivity, specificity, and accuracy were 93.2%, 100%, and 93.4%, respectively. The diagnostic accuracy could be increased to 96.4% in 2016–2018 (after the learning curve period). The overall complication rate was 8.6%. Neither life-threatening major complications, nor seeding through the biopsy tract, were observed. Our study supported the hypothesis that CT-CNB could be a complementary option for diagnostic tissue acquisition in patients with unresectable or metastatic pancreatic tumors when EUS-TA is either unsuitable or unavailable.

## 1. Introduction

Pancreatic cancer was the fourth and third leading cause of cancer mortality in the European Union and the United States, respectively, in 2018, and is predicted to become the second greatest cause of cancer mortality in Western countries by 2030 [1,2,3]. Pancreatic ductal adenocarcinoma (PDAC), which accounts for more than 85% of pancreatic malignancies, is a highly lethal disease, with a median overall survival of less than one year to a great extent due to 85–90% of patients presenting with locally advanced or metastatic diseases upon diagnosis, precluding curative intent surgery [4]. With the emergence of more effective combination chemotherapy regimens, such as FOLFIRINOX and nab-paclitaxel plus gemcitabine, conversion surgery following induction chemotherapy has been reported to improve the clinical outcomes of patients with either borderline resectable or initially unresectable PDAC, and has become a favorable treatment strategy in this particular group of patients. Definitive tissue diagnosis before commencing cancer treatment is the number one rule in oncology. However, the acquisition of pancreas tumor tissue has been challenging, due to its special anatomic position [5]. 

Ultrasound-guided, percutaneous fine needle aspiration (FNA) for intra-abdominal lesions has been used since the 1970s; yet, some pancreatic lesions are difficult to visualize on percutaneous ultrasounds due to bowel gas shadows [6]. Endoscopic, ultrasound-guided fine needle aspiration (EUS-FNA) was developed in the 1990s to overcome this limitation, becoming the mainstay in diagnosing pancreatic tumors. However, EUS-FNA, mostly in combination with rapid on-site evaluation (ROSE), provided only cytological specimens without tissue architecture, which may occasionally jeopardize accurate diagnoses. Currently, microcore capable needles, which were developed to obtain tissue samples for both cytology and histology examinations, have recently been widely applied in endoscopic, ultrasound-guided tissue acquisition (EUS-TA). These biopsy needles obtained very satisfactory results, also in the absence of ROSE [7].

It has been reported that EUS-TA is superior to ultrasound or computed tomography (CT)-guided percutaneous biopsies in terms of diagnostic accuracy and its relatively low risk of peritoneal seeding [8,9]. In a meta-analysis including 2761 patients, the sensitivity, specificity, and diagnostic accuracy of EUS-TA for pancreatic tumors were 90.8%, 96.5%, and 91.0%, respectively [5]. However, the high diagnostic accuracy of EUS-TA was established by the cooperation of highly experienced endoscopists and pathologists with ROSE. The diagnostic accuracy of EUS-TA might drop to 81% in the absence of ROSE, especially in a learning curve setting or in low-volume centers [10,11,12]. To perform ROSE, it’s necessary to have a pathology department near the endoscopy room with on-call pathologists during the EUS procedure, an arrangement which is not widely available, even in medical centers, therefore limiting the application of EUS-TA. A recent, randomized, controlled trial showed that the use of a 20-gauge needle could achieve a diagnostic accuracy of 87%, as compared to 78% of EUS-TA with a 25-gauge needle [13]. In that study, ROSE was only applied in 9% of cases in the 20-gauge needle group. The role and cost-effectiveness of ROSE in EUS-TA with large needles may require further evaluation. The facilities for the EUS procedure are not always available in every hospital; therefore, an alternative for pathological diagnosis of pancreatic lesions is warranted.

Percutaneous computed tomography-guided core needle biopsy (CT-CNB) make it possible to use of a large needle (generally 18-gauge) to get more satisfactory tissue samples. In the era of modern precision medicine, biopsy tumor tissue can serve not only for histologic diagnoses, but also for molecular profiling studies, such as next-generation sequencing. In one recent report, the success rates of sequencing were 100% with percutaneous core biopsy samples of metastatic lesions, versus those of 42.9% to 70.4% with EUS-guided FNA or FNB samples, mainly from pancreas [14]. 

Several techniques such as trans-hepatic, trans-gastric, hydrodissective, and pneumodissective approaches have been developed to avoid damage to visceral organs [15,16,17,18]. Over the past 10 years, our institute has developed a fat-transversing route approach for pancreatic biopsies which is able to reach most tumors in the pancreas without penetrating organs or vessels. The safety and efficacy of CT-CNB through this approach in a small cohort of 122 patients with a 96.7% success rate and 4.1% complication rate has been previously described [19]. Herein, we report the updated results in 420 patients.

## 2. Materials and Methods

### 2.1. Patients

In this retrospective cohort study, we identified a total of 482 patients who underwent CT-CNB of pancreatic lesions at National Cheng Kung University Hospital (NCKUH) from March 2004 to December 2018; 62 patients whose CT-CNB procedure did not use the coaxial needle system were excluded. As a result, a total of 420 patients who underwent coaxial CT-CNB of pancreatic lesions were included in current study. Patient demographics and clinical characteristics including age, sex, and tumor location and size were recorded, as well as the pathology report plus both the clinical and radiological follow-ups. All patients gave their written informed consent before undergoing this invasive procedure; furthermore, the retrospective clinical data collection was approved by the NCKUH Institutional Review Board (No. A-ER-108-113), and followed the Declaration of Helsinki.

### 2.2. Computed Tomography-Guided Core Needle Biopsy Technique

Complete blood count (CBC), prothrombin time, and partial thromboplastin time were checked within three days before procedure. Each patient was accessed by three well-experienced, interventional radiologists to determine the proper route of the biopsy. Six major fat-transversing routes have been previously developed at our institute (Figure 1) [19]. 

A demonstrative case of CT-CNB via the detour route 6 (DR6) is shown in Figure 2. Other detour routes (DR1 to DR5) are shown in Appendix A
Figure A1, Figure A2, Figure A3, Figure A4 and Figure A5. Figure A3 also demonstrates the potential application of posterior DR3 to DR6 approaches in patients with massive ascites.

First, 100 mg of Tramadol was applied intravenously 30 min before the procedure, without routine sedation in most cases. Local anesthesia of 2 mL with 2% xylocaine was injected subcutaneously at the needle entry point. Under CT guidance, a 17-gauge coaxial needle, which served as an outer sheath, was inserted into the target lesion along a curved route consisting of one to three short, straight-line sections. The location of the sheath needle was confirmed by non-contrast CT scan; then, an 18-gauge needle was inserted to obtain the tissues (18 G Tru-core needle (15–20 cm); Angiotech, Vancouver, CA, USA). Several biopsies could be obtained through the coaxial needle system without repositioning the guide needle, preventing tumor seeding along the biopsy tract. Generally, one to three core biopsy specimens were collected and sent to the Department of Pathology for routine histopathologic work-up including hematoxylin & eosin (H&E) stain and immunohistochemistry stain, when necessary. Blood pressure, heart rate, oxygen saturation, and electrocardiogram were monitored continuously throughout the procedure. In those with significant bleeding observed from the coaxial needle tract during the procedure, gelfoam was used for tract embolization with stable hemostasis. A non-contrast CT scan was performed following the biopsy to determine if there were any immediate complications. 

### 2.3. Statistical Analysis

There would be three categories of diagnosis: neoplasms, benign lesions, and inconclusive cases. The category of “neoplasms” included definitive malignancies, such as adenocarcinoma, lymphoma, or neuroendocrine tumors, as well as borderline malignant neoplasm like intraductal papillary mucinous neoplasm (IPMN). “Benign lesion” described samples with benign diagnoses for at least six months afterwards, to make sure the disease was truly benign. “Inconclusive cases” were those with uncertain diagnoses with less than three months of follow-up periods. Continuous variables were presented as mean ± standard deviation (SD) or median with range, whereas categorical variables were expressed as numbers or percentages. All statistical analyses were performed using R 3.6.0 (R Foundation for Statistical Computing, Vienna, Austria).

## 3. Results

### 3.1. Patient Characteristics

The overall success rate for tissue acquisition of the fat-traversing route technique was 99.3%. Patient demographics and clinical characteristics are summarized in Table 1. The overall median procedure time, as calculated from the first CT scan to the removal of the needle, was 28 min (range 12–54 min), including 15–20 min to advance the 17-gauge outer sheath to the target lesion. The procedure time was affected by the tumor size (Table 1) and location, but not by the approaching route (Appendix A
Table A1). The final diagnosis was neoplasms in 395 cases; the primary tumor size and locations, final histology diagnosis, and TNM staging are summarized in Table 2.

### 3.2. Final Diagnosis and Diagnostic Performance

The designation of “true positive” was given to those diagnosed as “positive for neoplasm”, as confirmed by surgical specimens, or to those developing significant disease progression during follow-up. “True negative” referred to those with a benign diagnosis matching the benign clinical course for at least six months of follow-up. “False negative” was designated for those having a benign diagnosis, but who then proved to be malignant by other specimens or by the development of clinically-significant metastases during follow-up. Eleven patients with uncertain depositions with follow-up periods of less than three months were excluded from diagnostic performance analyses. The overall sensitivity, specificity, and accuracy of the technique were 93.2%, 100%, and 93.4%, respectively. The diagnostic accuracy slightly increased from 91.1% in tumors <20 mm to 93.4% in tumors >41 mm. For false negative cases, the final diagnosis was confirmed by biopsies from metastatic lesions (Table 3). After the period of learning curve, the overall diagnostic accuracy significantly increased from 90.7% in 2004–2015 to 96.4% in 2016–2018 (Figure 3).

### 3.3. Safety Profiles

The overall complication rate was 8.6%, with mainly local hematoma (3.8%) or intra-procedural tract bleeding (3.8%), as shown in Table 4. Despite the procedure being undertaken by experienced radiologists, 7 out of 420 patients had complications of trans-organ punctures, though no life-threatening, major events were observed. The complication and trans-organ rates were not significantly correlated with body mass index (BMI). All complications were manageable by conservative treatment without major bleeding in cases with trans-renal procedures. There was only one case requiring hospitalization for complication management; a 74-year-old female with chronic hepatitis B-related liver cirrhosis and ascites developed peritonitis three days after CT-CNB, but fully recovered after treatment with antibiotics.

Among 5 patients with mucinous neoplasm, two had operable tumors and underwent curative surgery within one month after CT-CNB. One patient developed peritoneal carcinomatosis 11.7 months after CT-CNB, while the other showed no evidence of peritoneum recurrence during a 17.5-month follow-up. Of the other three patients with unresectable mucinous neoplasm, one patient lost to follow-up, while the other two patients had no radiological evidence of peritoneal carcinomatosis during a follow-up of 6.2 months and 9.1 months, respectively.

Of the 395 patients with confirmed malignancies, 27 (6.8%) had clearly operable tumors (TNM stage I and II), while 93 (23.5%) had locally-advanced, unresectable tumors (stage III) and 275 (69.6%) patients had metastatic diseases (stage IV) initially (Table 5). Generally, follow-up CT scans were performed every 3 months during treatment, according to the regulations of the National Health Insurance in Taiwan. After the exclusion of 69 patients who had peritoneal carcinomatosis at diagnosis, and 67 patients who were lost to follow-up, the median duration of CT scan follow-ups in the remaining 259 patients was 13.4 months for stage I, 10.2 months for stage II, 8.8 months for stage III, and 6.1 months for stage IV. The incidences of developing peritoneal metastasis were 16.7% in stage I, 20% in stage II, 35.7% in stage III, and 21.5% in stage IV. The median times from biopsy to detection of peritoneal carcinomatosis were 11.7 months in stage I, 6.1 months in stage II, 10.0 months in stage III, and 7.8 months in stage IV, as shown in Table 5.

## 4. Discussion

Nowadays, EUS-TA is considered to be superior to CT-FNA, because of its better diagnostic accuracy and lower risk of developing peritoneal carcinomatosis by the feasibility of resecting the EUS-TA tract during subsequent surgery; thus, its use has been recommended for pancreatic tumor tissue acquisition in practice guidelines [20,21]. However, the guidelines asserting that better diagnostic accuracy was obtained with EUS-TA compared to the percutaneous approach were primarily based on studies comparing EUS-TA to CT-FNA, rather than CT-CNB. Currently, coaxial CT-CNB is more widely used in CT-guided, percutaneous approaches, with a satisfactory diagnostic accuracy of 82.2–98.1% (Table 6). There was no head-to-head study to compare the diagnostic accuracy and complication rates between EUS-TA and CT-CNB.

It has been reported that the accuracy of EUS-TA improves according to tumor size. A recent study demonstrated that the accuracy was up to 91.7% in tumors <10 mm and 98.7% in tumors >40 mm [22]. In a current study, the accuracy of CT-CNB was shown to have significantly increased, up to 94.4% and 97.7% in tumors ranging from 10 to 20 mm and 20 to 40 mm, respectively, after the learning curve period. However, CT-CNB could be difficult for pancreatic lesions <10 mm. To the best of our knowledge, our study is the largest cohort of coaxial CT-CNB for pancreatic lesions in the extant literature. The current results suggest that the diagnostic performance of CT-CNB can be non-inferior to standard EUS-TA when tumor sizes are greater than 10 mm. CT-CNB could serve as an alternative if EUS-TA is either unsuitable or unavailable, especially in unresectable or metastatic malignancies.

Seeding through the biopsy tract is one of the major concerns of percutaneous biopsies. However, PDAC is frequently diagnosed in late stages, with incidences of peritoneal carcinomatosis ranging from 9.1% to 32% at diagnosis [30,31,32]. Therefore, the development of peritoneal carcinomatosis may be a natural course, rather than procedure-related. NCCN guidelines assert a lower frequency of peritoneal carcinomatosis of EUS-FNA, which was primarily based on two studies [9,33]. One featured 197 patients (72 EUS-FNA and 125 percutaneous FNA); the incidence of peritoneal carcinomatosis was 0% and 2.4%, respectively, which was not statistically different (Table 7). The other study reported a statistically higher incidence of peritoneal carcinomatosis in percutaneous FNA (16.3%) than in EUS-FNA (2.2%). However, both studies used percutaneous-FNA as a comparison, rather than the coaxial core needle biopsy. On the other hand, it has been reported that the incidence of peritoneal seeding does not increase by pre-operative CT-FNA [34]. Another study demonstrated that the risk of peritoneal seeding by percutaneous US-FNA was similar to that by EUS-FNA [35]. The percutaneous FNA that directly penetrates the tumor without a protective outer sheath is theoretically more prone to needle tract tumor cell seeding. Of the 27 patients with stage I and II diseases in the current study, 3 of 21 (14.3%) patients who underwent curative surgery after CT-CNB developed peritoneal recurrence during follow-up; for the remaining 6 patients who did not receive surgery, 2 (33.3%) had peritoneal carcinomatosis during follow-up. The 18.5% (5/27) overall incidence of peritoneal carcinomatosis in patients with stage-I and -II diseases in our study was consistent to that in the JSPAC-01 study, in which peritoneum was recognized as the site of first recurrence, either alone or along with other metastatic site(s), in 30 out of 190 (15.8%) R0/R1 resected pancreatic cancer patients receiving adjuvant gemcitabine therapy [36]. The results concur with the report of Maturen et al. that the coaxial needle system has lower risk of biopsy tract seeding, as the outer sheath could prevent this via the biopsy tract [37].

In our study, we identified 12 cases with peritoneum-abdominal, wall-skin axis metastasis alone, without visceral organ involvement, including 7 cases of solitary peritoneal seeding only, 2 cases of both peritoneum and sister Joseph’s node, one case of both abdominal wall and peritoneum, one case with both skin and sister Joseph’s node, and one case of sister Joseph’s node only. Interestingly, none of these patients had radiologic evidence to show definitive tumor implantation along previous the needle puncture site and biopsy tract. Of the only previous study that reported the incidence of peritoneal carcinomatosis after CT-CNB with the coaxial approach, Tyng et al. reported none of their 103 patients had such complications; the same was true of our patients (Table 6). Our experience suggests that CT-CNB is a workable method with low risk of biopsy tract seeding using the coaxial system. However, since there are still concerns of biopsy tract seeding and incidental penetration of neighboring organs, currently, CT-CNB should only be applied for patients with unresectable or metastatic diseases in the absence of EUS facilities.

There were three limitations in our study. First, there was no pre-specified protocol for follow-up of peritoneal carcinomatosis. Since the occurrence of ascites or peritoneal carcinomatosis is common in advanced stage-III and -IV pancreatic cancer, it was difficult to determine whether new, radiologically-detectable, peritoneal carcinomatosis and/or ascites were procedure-related or just the disease course per se. Therefore, we can only discuss and compare the incidence of peritoneal carcinomatosis of current stage-I and -II patients who underwent surgical resection after CT-CNB with that in a JASPAC-01 trial of adjuvant S-1 vs. gemcitabine in curatively-resected pancreatic cancer. Second, EUS-TA was not available in our institute until 2017, so we were not able to compare the diagnostic performance and safety profiles in different imaging modalities in our institute. Finally, CT-CNB is a highly technical procedure, and the current report is from a single, high-volume medical center with experienced radiologists.

## 5. Conclusions

Our study demonstrated that CT-CNB, a previously widely applied method for the diagnosis of pancreatic lesions, has an excellent technique success rate of 99.3%, a diagnostic accuracy of up to 96.4%, and a manageable complication rate of 8.6% in a high-volume medical center with experienced radiologists. Nowadays, EUS-TA is the preferred diagnostic procedure for pancreatic tumors due to its better safety profiles; however, our study suggests that CT-CNB could be a complementary option for patients with unresectable or metastatic diseases in the absence of EUS facilities. Further prospective trials to elucidate the role of CT-CNB in the era of available EUS-TA are warranted.

## Figures and Tables

**Figure 1 jcm-08-01633-f001:**
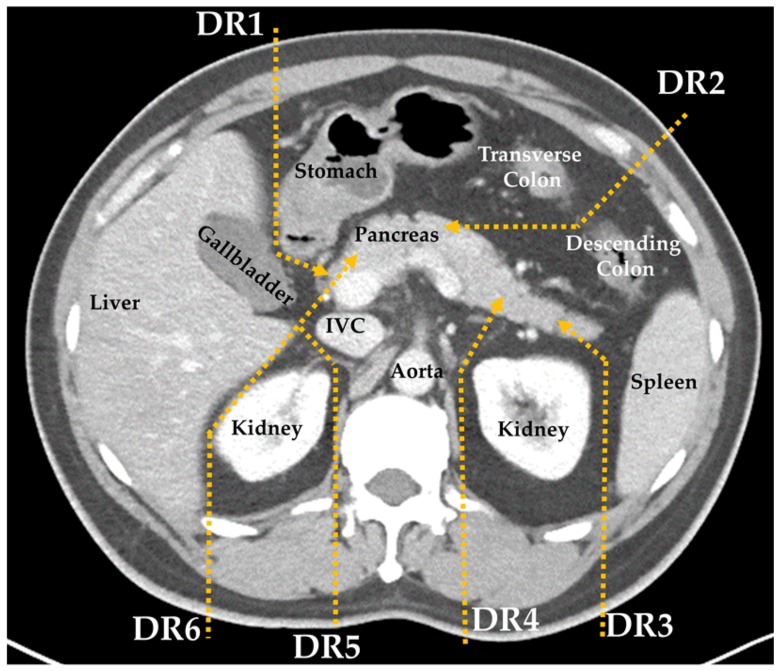
Diagram of the six major fat-transversing routes: Detour Route 1 (DR1) to pass between the stomach and gallbladder for head lesion; DR2 to pass through the splenic flexure space between the transverse and descending colon for body lesions; DR3 and DR4 to bypass the left kidney and spleen for body and tail lesions; and DR5 and DR6 to bypass the liver, right kidney, and inferior vena cava (IVC) for head or uncinate process lesions.

**Figure 2 jcm-08-01633-f002:**
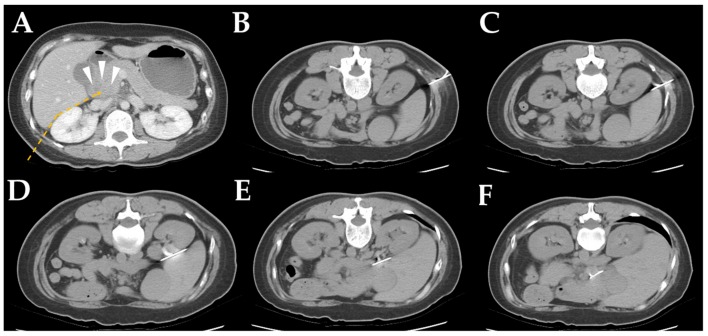
Computed tomography (CT)-guided, core needle biopsy images via detour route 6 in a 51-year-old female with a pancreatic head tumor. (**A**) Contrast-enhanced CT demonstrated a hypodense tumor in the pancreatic head (arrowhead). The detour route was planned to avoid penetration of the liver and kidney (dotted line). (**B**,**C**) A prone non-enhanced CT scan showed the insertion path of the coaxial guiding needle through the fat between the liver and kidney. (**D**,**E**) When the needle reached the planned turning point, the needle direction shifted slightly to avoid penetrating of the liver. (**F**) After successful insertion of the coaxial needle into the lesion, the biopsy gun was fired into the pancreatic head tumor.

**Figure 3 jcm-08-01633-f003:**
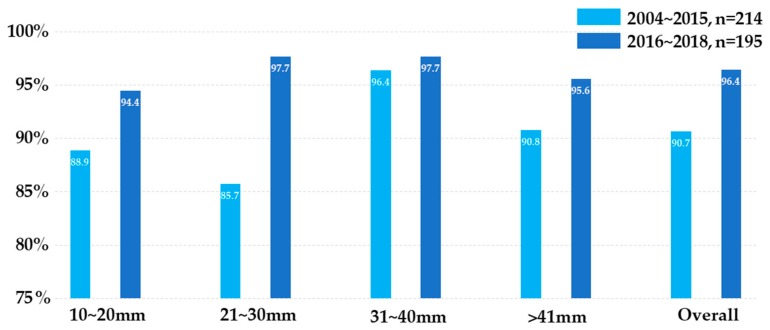
Diagnostic accuracy categorized by tumor size. The light blue bar represents cases performed between 2004 and 2015, while the dark blue represents cases after 2016. The diagnostic accuracy significantly improved after the learning curve period and with experience.

**Table 1 jcm-08-01633-t001:** Patient demographics and clinical characteristics.

Tumor Size	10–20 mm *n* = 47	21–30 mm *n* = 103	31–40 mm *n* = 99	>41 mm *n* = 171	Overall *n* = 420
Age, mean ± SD, year	64.2 ± 12.0	63.9 ± 11.5	63.0 ± 12.7	60.7 ± 11.4	62.4 ± 11.9
Gender (male/female)	26/21	57/46	60/39	105/66	248/172
Location					
Head	37	71	46	59	213 (50.7%)
Body	6	27	38	59	130 (31.0%)
Tail	4	5	15	53	77 (18.3%)
Access					
Detour route 1	26	49	43	51	169 (40.2%)
Detour route 2	16	32	39	82	169 (40.2%)
Detour route 3	2	4	3	17	26 (6.2%)
Detour route 4	0	2	6	11	19 (4.5%)
Detour route 5	3	10	6	7	26 (6.2%)
Detour route 6	0	6	2	3	11 (2.6%)
Procedure time, median (range), minutes	34 (14–54)	31 (18–53)	30 (22–46)	23 (12–42)	28 (12–54)
Number of tissue strips, median (range)	2 (1–3)	2 (1–3)	2 (1–3)	2 (1–5)	2 (1–5)
Performed year					
2004–2015	28	58	55	77	218 (57.9%)
2016–2018	19	45	44	94	202 (42.1%)
Technique performance					
Success tissue acquisition	47	103	99	168	417 (99.3%)
Inadequate specimen	0	0	0	3	3 (0.7%)
Final diagnosis					
Neoplasms	41	97	95	162	395 (94.1%)
Benign lesions	4	2	3	5	14 (3.3%)
Inconclusive cases *	2	4	1	4	11 (2.6%)

* Inconclusive cases were those with uncertain diagnoses and follow-up periods of less than three months.

**Table 2 jcm-08-01633-t002:** Characteristics of 395 confirmed pancreatic neoplasms.

	Positive Diagnosis by CT-CNB/All Cases, *n* (%)	Size, Median (Range), cm	Stage			
			I *n* = 6	II *n* = 21	III *n* = 93	IV *n* = 275
Histology type						
Adenocarcinoma	302/323 (93.5%)	3.5 (1.1–15.5)	3	19	84	217
Poorly differentiated carcinoma	13/13 (100%)	4.7 (2.5–9.7)	0	0	3	10
Metastatic tumor	15/17 (88.2%)	3.6 (1.6–6.7)	0	0	0	17
Neuroendocrine tumor	17/17 (100%)	4.0 (2.0–17.0)	1	1	3	12
Mucinous neoplasms	5/5 (100%)	4.2 (1.7–7.0)	1	1	1	2
Lymphoma	9/10 (90.0%)	5.2 (2.9–9.1)	1	0	0	9
Adenosquamous carcinoma	5/6 (83.3%)	4.2 (3.4–8.0)	0	0	1	5
Others *	2/4 (50.0%)	4.8 (1.7–8.5)	0	0	1	3
Location						
Head	179/194 (92.3%)	3.0 (1.1–9.0)	5	17	56	116
Body	120/127 (94.5%)	3.9 (1.6–17.0)	1	3	33	90
Tail	69/74 (93.2%)	4.8 (1.8–15.5)	0	1	4	69

* One case of acinar cell carcinoma, one case of pancreatoblastoma and two cases without pathology confirmation. CT-CNB, Percutaneous computed tomography-guided core needle biopsy.

**Table 3 jcm-08-01633-t003:** Diagnostic performance of CT-CNB by tumor size in 409 cases with conclusive diagnosis.

Tumor Size	10~20 mm *n* = 45 *	21~30 mm *n* = 99 *	31~40 mm *n* = 98 *	>41 mm *n* = 167 *	Overall *n* = 409 *
True positive	37	82	92	151	368
True negative	4	2	3	5	14
False positive	0	0	0	0	0
False negative	4	9	3	11	27
Sensitivity	90.2%	90.7%	96.8%	93.2%	93.2%
Specificity	100%	100%	100%	100%	100%
Accuracy	91.1%	90.9%	96.9%	93.4%	93.4%
Subsequent diagnostic procedures in first CT-CNB false negative cases					
Re-biopsy †	0	3	0	3	6
Biopsy via EUS	0	0	0	1	1
Biopsy of metastatic lesion	3	2	0	6	11
Surgical specimen	1	2	2	0	5
Clinically significant metastases	0	0	1	1	2

* Inconclusive cases lacking a minimum follow-up of three months were excluded. † Including those receiving a repeat biopsy right after the first biopsy or during follow-up.

**Table 4 jcm-08-01633-t004:** Safety profiles.

	BMI < 18.5 *n* = 35	BMI 18.5–22.9 *n* = 175	BMI 23–24.9 *n* = 91	BMI ≥ 25 *n* = 119	Overall *n* = 420
No complications	35 (100%)	161 (92.0%)	85 (93.4%)	103 (86.6%)	384 (91.4%)
Major complications	0	0	0	0	0
Minor complications					
Local hematoma	0	7	5	4	16
Intra-procedural tract bleeding	0	6	0	10	16
Pancreatitis	0	0	0	1	1
Transient hypotension	0	1	0	1	2
Bacteria peritonitis	0	0	1	0	1
Trans-organ					
Trans-gastric	0	1	3	2	6
Trans-renal	0	1	0	0	1

**Table 5 jcm-08-01633-t005:** Incidence of peritoneal carcinomatosis at diagnosis and during follow-up.

	Stage I *n* = 6	Stage II *n* = 21	Stage III *n* = 93	Stage IV *n* = 275
PC before CT-CNB	0	0	0	21/48 *
No PC before CT-CNB	6	21	93	206
Loss of F/U after CT-CNB, *n*	0	1	9	57
F/U image available, *n*	6	20	84	149 †
Median (range) F/U duration, months	13.4 (7.0–16.3)	10.2 (0.5–41.9)	8.8 (0.1–90.3)	6.1 (0.2–99.0)
PC detected during follow-up, *n* (%) PC alone PC and other sites of metastasis	1 (16.7%) 0 1	4 (20%) 1 3	30 (35.7%) 15 15	32 (21.5%) 18 14
Median (range) time to PC, months	11.7	6.1 (4.3–10.6)	10.0 (2.1–28.9)	7.8 (1.5–24.4)

PC: peritoneal carcinomatosis; CT-CNB: computed tomography-guided core needle biopsy; F/U: follow-up. ***** 21 patients had PC alone; 48 patients had PC and other sites of metastases; † Excluding those with peritoneal carcinomatosis at diagnosis.

**Table 6 jcm-08-01633-t006:** Diagnostic accuracy and complication of percutaneous biopsy using coaxial needle system.

Reference	Guidance Image Modality	Case Number Coaxial System/All cases	Accuracy	Complications	Biopsy Tract Seeding
[23]	CT	29/29	82.2%	13.8%	Not mentioned
[24]	CT/US	110/110	94.4%	2.7%	Not mentioned
[25]	CT/MRI	30/30	93%	16.7%	Not mentioned
[26]	US	75/88	93.2%	3.3%	Not mentioned
[27]	CT	103/103	98.1%	8.7%	0
[28]	MRI	31/31	93.5%	6.5%	Not mentioned
[29]	CT/US	67/82	82.9%	11.0%	Not mentioned
Current study	CT	420/420	93.4%	8.6%	0

**Table 7 jcm-08-01633-t007:** Studies comparing tumor seeding between EUS-FNA and percutaneous-FNA.

Reference		EUS-FNA	Percutaneous-FNA	*p*-Value *
Micames C, et al. [9]	*N*	46	43	
	Tumor seeding, *n* (%)	1 (2.2%)	7 (16.3%)	0.03
Okashi HH, et al. [31]	*N*	72	125	
	Tumor seeding, *n* (%)	0%	3 (2.4%)	0.3
Matsuyama, M, et al. [35]	*N*	75	46	
	Tumor seeding, *n* (%)	0%	0%	1.0

* Calculated by Fisher exact test.

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
