# Peer review of "Percutaneous Computed Tomography-Guided Coaxial Core Biopsy for the Diagnosis of Pancreatic Tumors"

_jcm, 2019, doi:10.3390/jcm8101633_

Round 1

Reviewer 1 Report

In the methods section, please elaborate on how you are able to take a straight coaxial needle along a curved or bent path without injuring adjacent organs or bowel.  The average reader will not believe that you are able to take a straight 17 gauge coaxial needle along a 45 degree turn without hydrodissection.  To me these routes look very risky to reach some portions of the pancreas.

Please show intrabiopsy needle positioning example CT images of each of the DR1 through DR6 routes to prove that you can take these courses safely.

Also, please quantify in your 502 procedures, how many biopsies used each DR1 through DR6 routes.  This could be shown in a chart. This is necessary to show readers the types of procedures you performed.

It would be helpful to have the BMI info of these patients as larger visceral fat would allow safer biopsy in all patients.

Additionally, minor English errors should be corrected.

Author Response

Dear reviewer,

Thank you for your meaningful suggestion!

Intra-abdominal fat between the GI is soft and is easy to change the direction of the needle. Therefore, it is not difficult to succeed except for uncooperative patients. The coaxial needle is slightly blunt, so it is not easy to hurt bowel. At least, this has not been occurred in our experience. In the most cases, there are not only one way to the pancreatic mass from abdominal wall or back. We choice the most simple and safe way to puncture. For example, we are usually used DR5 for the uncinate process of the pancreatic mass. We demonstrated the real practice image in each detour route in Fig. 2 and Fig. A1-A5. We already added the information of DR1 to DR6 in table 1. By the way,
in order not to confuse reader, we modified our results of "502 procedures" into "482 patients". The information of BMI was added in Table 4.

Reviewer 2 Report

Dear Authors,

I read with interest your papers and I thank you for your efforts. Nonetheless my comments require major changes.

Introduction.

EUS-guided FNA has left its plasce to the definition of EUS-guided tissue acquisition (which includes FNA-FNB and associated tecniques). EUS_TA is the term I would choose to discuss this topic.

In line 54 you state that EUS-TA requires sedation (but you can also perform the procedure on mild sedation)t. I think also percutaneous biopsy requires some level of sedation.

While guidelines still recommend ROSE, more recent evidences form the literature show how EUS-TA by FNB  requires a limited number ofpasses without ROSE and with very satisfactory results (among others: A multicenter randomized trial comparing a 25-gauge EUS fine-needle aspiration device with a 20-gauge EUS fine-needle biopsy device.van Riet PA, Larghi A, ..Bruno MJ.Gastrointest Endosc. 2019 Feb;89(2):329-339. )

Results: 

the mean diameter of the lesion is relatively large (4.2 cm). A large diameter is associated with the best performance for EUS-TA,nearly 100%. comparison of your results should be towards lesion of similar diameter. otherwise you should distinguish among different lesion size (<10 mm, 10-20 mm, etc..).

In comparison with the literature for EUS-TA the false negative rate is high, as well as the complication rate. this a problem expecially in surgical candidates patients.

Considering all these points I have to conclude that your approac should be applide to paitents with stage III-IV  (not surgical candidates).

Moreover the conclusion about the incidence of peritoneal carcinosis cannot be derived from these results.

I think that this question requires a prospective observational study. As an alternative a retrospective study may offer some information but included patients should be  well categorixed as stage I-II-III or IV at baseline. patients with stage III or IV are at high risk of misdiagnosed carcinosis so no wonder that the authors find no statistically significan difference in carcinosis after their procedure. 

Anyway the authors offer a large retrospective study of percutaneous biopsy. In my opinion this tecnique could find a placein diagnostic management of pancreatic lesions in non surgical patients.so I suggest reevalutation after major revision

bet regards

Author Response

Dear reviewer,

Thank you for your meaningful suggestion!

EUS-TA is really more suitable than EUS-FNA/FNB. We had modified our terms in the article. Sedation was not routinely administered in most cases for percutaneous biopsy in our hospital except for uncooperative patients. In general, we prefer not to do biopsy in uncooperative patients rather than performing biopsy under sedation. Thank you for the updated information. We already added this information into introduction section in line 73. We totally agree with you that percutaneous biopsy should only be applied to paitents with stage III-IV. We added this conclusion in line 256. We also agree with you completely that the issue of peritoneal carcinomatosis requires a prospective trial. Therefore, we revised our statement and only displayed the results of carcinomatosis in different stage in Table 5 as your suggestion.

Round 2

Reviewer 1 Report

In Table 1 description please clarify what the size measurements on the top header row are referring to (tumor size).  You did this fine in table 3.

Please add a one sentence explanation in figure 3 caption why there improved improved diagnostic accuracy in 2016-2018 versus 2004-2015.  this is helpful for the reader to quickly understand what you are trying to present. Was this just better experience by the operator or was there a fundamental change that occurred between these times.

Otherwise, great changes and the extra figures and images are appreciated and help to really show the reader your technique and experience in this type of procedure.

Author Response

Dear reviewer,

Thank you very much for the recommendation which makes the article more reliable and of higher quality.

The missing word "tumor size" in Table 1 is corrected. The device is actually the same, so we think the improvement  of accuracy is as you mentioned,  just better experience by the operator. We add a sentence in Figure 3.

Reviewer 2 Report

Dear Authors, 

I read your revisions. For clarity to the reader I would suggest some further elucidations.

line 59: "..However, EUS-guided FNA, mostly in combination with ROSE, .."

line 60-62 "Currently microcore capable needles have been developed to obtain tissue samples, both ofr cytology and histolgy examinations and have been widely applied. These FNB-needles obtained very satisfactory results also in absence of ROSE (Histologic retrieval rate of a newly designed side-bevelled 20G needle for EUS-guided tissue acquisition of solid pancreatic lesions.United European Gastroenterol J. 2019 Feb;7(1):96-104). 

I would suggest these changes since the role of ROSE has deeply decreased since the introduction of third generation needles for EUS-TA (Histologic retrieval rate of a newly designed side-bevelled 20G needle for EUS-guided tissue acquisition of solid pancreatic lesions.United European Gastroenterol J. 2019 Feb;7(1):96-104).

Line 69, ..(ROSE), expecially in a learning curve setting or in low volume centers

(Wani S, Mullady D, Early DS, et al. The clinical impact of immediate on-site cytopathology evaluation during endoscopic ultrasound-guided fine needle aspiration of pancreatic masses: A prospective multicenter randomized controlled trialAm J Gastroenterol 2015; 110: 1429–1439

Iglesias-Garcia J, Dominguez-Munoz JE, Abdulkader I, et al. Influence of on-site cytopathology evaluation on the diagnostic accuracy of endoscopic ultrasound-guided fine needle aspiration (EUS-FNA) of solid pancreatic massesAm J Gastroenterol 2011; 106: 1705–1710). 

line 78-79: EUS FNB or other diagnostic procedure may not be indicated if the patient has such a serious condition to make sedation a risk, so I would suggest removing this comment

line 148: I guess the term is borderline malignant.

In Discussion:

line 215: change EUS-FNA with EUS-TA

line 216: comment: the advantage of EUS-TA is the possibility to resect surgically the tract of potential seeding,. which can't be done in percutaneous approach (US or TC guided)

line 224: I would suggest to remove : (47% ofr tumous <..>40) since in the following sentence similar data are reported (and they have a reference).

final comment: although relatively mild the complications ot percutaneous approach are still hige when compared to those obatined by EUS-TA, so this tecnique should be suggested only in absence of EUS facilities.

I thank the authors for their work and I Hope their paperwill be accepted after these minor changes

regards

Author Response

Dear reviewer,

Thank you very much for your constructive feedback and expert editing, which makes the article more reliable and of higher quality.

The sentence in line 59 is revised. The original sentence in line 60-62 and reference are revised and now in line 61-64. The original sentence in line 69 and reference are revised and now in line 72. The sentence in 78-79 is removed. The typing error in line 148 is revised. "EUS-FNA" in line 215 is corrected to "EUS-TA". This precious comment is added in line 219-220 now. The original sentence in line 224 is modified and now in line 228. We totally agree with your recommendation that CT-CNB should be suggested only in the absence of EUS facilities.  We add this conclusion in line 261 and 278.